# Fooling Contrastive Language-Image Pre-trained Models with CLIPMasterPrints

**Matthias Freiberger**                                                      *mafr@di.ku.dk*
*Copenhagen University*

**Peter Kun**                                                                 *peku@itu.dk*
*IT University of Copenhagen*

**Christian Igel**                                                            *igel@di.ku.dk*
*Copenhagen University*

**Anders Sundnes Løvlie**                                                     *asun@itu.dk*
*IT University of Copenhagen*

**Sebastian Risi**                                                            *sebr@itu.dk*
*IT University of Copenhagen*

**Reviewed on OpenReview:** *https://openreview.net/forum?id=ZFZnvGXXMm*

## Abstract

Models leveraging both visual and textual data such as Contrastive Language-Image Pre-training (CLIP), are the backbone of many recent advances in artificial intelligence. In this work, we show that despite their versatility, such models are vulnerable to what we refer to as fooling master images. Fooling master images are capable of maximizing the confidence score of a CLIP model for a significant number of widely varying prompts, while being either unrecognizable or unrelated to the attacked prompts for humans. The existence of such images is problematic as it could be used by bad actors to maliciously interfere with CLIP-trained image retrieval models in production with comparably small effort as a single image can attack many different prompts. We demonstrate how fooling master images for CLIP (CLIPMasterPrints) can be mined using stochastic gradient descent, projected gradient descent, or blackbox optimization. Contrary to many common adversarial attacks, the blackbox optimization approach allows us to mine CLIPMasterPrints even when the weights of the model are not accessible. We investigate the properties of the mined images, and find that images trained on a small number of image captions generalize to a much larger number of semantically related captions. We evaluate possible mitigation strategies, where we increase the robustness of the model and introduce an approach to automatically detect CLIPMasterPrints to sanitize the input of vulnerable models. Finally, we find that vulnerability to CLIPMasterPrints is related to a modality gap in contrastive pre-trained multi-modal networks. Code available at https://github.com/matfrei/CLIPMasterPrints.

## 1 Introduction

In recent years, contrastively trained multi-modal approaches such as Contrastive Language-Image Pre-training (CLIP; Radford et al., 2021) have increasingly gained importance and form the backbone of many recent advances in artificial intelligence. Among numerous useful applications, they constitute a powerful approach to perform zero-shot image retrieval, zero-shot learning and play an important role in state-of-the-art text-to-image generators (Rombach et al., 2022). Yet, recent work raises the question of robustness and safety of CLIP-trained models. For example, Qiu et al. (2022) find that CLIP and related multi-modal

Figure 1: Heatmap of CLIP-assigned cosine similarities of famous artworks and their titles, as well as a random noise baseline and our found CLIPMasterPrints for SGD, LVE and PGD approaches (marked with red frame) as returned by a pre-trained CLIP model. The mined fooling examples outperform all artworks in terms of CLIP score and can therefore fool the model for all targeted titles shown.

approaches are vulnerable to distribution shifts, and several research groups have successfully mounted adversarial attacks against CLIP (Noever & Miller Noever, 2021; Daras & Dimakis, 2022; Goh et al., 2021). In this paper we show for the first time that, despite their power, CLIP models are vulnerable towards fooling master images, or what we refer to as *CLIPMasterPrints*, and that this vulnerability appears to be closely related to a modality gap between text and image embeddings (Liang et al., 2022).

*CLIPMasterPrints* are capable of maximizing the confidence score of a CLIP model for a broad range of widely varying prompts, while for humans they appear unrecognizable or unrelated to the prompt. This ability can effectively result in the CLIPMasterPrint being chosen over actual objects of a class when being compared to each other by the attacked model. The existence of such images is problematic as it could be used by bad actors to maliciously interfere with CLIP-trained image retrieval models in production with comparably small effort as a single image can attack many different prompts. For instance, inserting a single CLIPMasterPrint into an existing database of images could potentially disrupt the system's functionality for a wide range of search terms, as for each targeted search term the inserted CLIPMasterPrint is likely to be the top result. This could exploited in malicious ways for censorship, adversarial marketing and disrupting the quality of service of image retrieval systems (for details see Section 5).

Consequentially, the existence of such images raises interesting questions on the efficacy and safety of multi-modal approaches to zero-shot image retrieval and possibly further applications as well.

Our contributions are as follows: we introduce fooling master images (CLIPMasterPrints) for contrastive multi-modal approaches and show that they can be mined using different techniques with different trade-offs:

(1) A stochastic gradient descent (SGD) approach, which is highly performant but requires knowledge of the model weights. (2) A blackbox optimization approach based on the family of *Latent Variable Evolution* (LVE; Bontrager et al., 2018; Volz et al., 2018) attacks, which does not have that limitation, but operates on a reduced search space and requires more iterations to achieve good results. (3) As both approaches can not be integrated into existing natural images, we also mine using a third approach based on projected gradient descent (PGD; Kurakin et al., 2016; Madry et al., 2018), which produces more natural-looking fooling images, but again requires the model's weights. While all three techniques have been used in variation in different contexts before, our approaches differ w.r.t. optimized loss function, nature of found solutions (on-manifold vs off-manifold), used generative model (LVE), and application domain. Furthermore, our emphasis in this work is on the results and insights we obtain, and their subsequent analysis: We find that the mined fooling examples tend to generalize to non-targeted prompts that are textually related to targeted prompts. In connection with the wide application of CLIP models and the fact that we find more recent similar approaches  Li et al. (2022); Zhai et al. (2023) to be vulnerable as well, this generalization effect adds to the gravity of the attack. We propose mitigation approaches focusing on the robustness of the model itself on the on hand as well as the detection of CLIPMasterPrints to enable sanitizing the model's inputs on the other hand. Finally, we demonstrate that mitigating the modality gap inherent to contrastive multi-modal models  (Liang et al., 2022) is also an effective counter-strategy to reduce the effectiveness of the mined fooling images. Consequentially, our results point towards a strong connection between a vulnerability to CLIPMasterPrints and a modality gap between text and image embeddings and thus opens up interesting future research directions, in finding even more effective mitigation strategies for both phenomena.

## 2  Related work

The notion of fooling examples was originally introduced by Nguyen et al. (2015), in which the authors generate fooling examples for individual classes for convolutional neural network (ConvNet) classifiers (LeCun et al., 1998) using genetic algorithms and compositional pattern producing networks (Stanley, 2007). In later work, Alcorn et al. (2019) showed that ConvNets can even be easily fooled by familiar objects in different and out-of-distribution poses. The main difference to our work is that the authors generate images that are misclassified as just one concrete class, while our images fool the network with respect to many classes or prompts.

Adversarial examples and adversarial learning (Chakraborty et al., 2018; Ozdag, 2018; Akhtar et al., 2021) are closely related to generating fooling examples, where usually adversarial examples can be disguised as regular images. The gradient-based approaches we apply in this paper are related to a number of popular gradient-based adversarial attacks, foremost the fast gradient sign and PGD methods (Goodfellow et al., 2015; Kurakin et al., 2018; 2016; Madry et al., 2018). Contrary to how these attacks are usually applied though, we optimize a loss function targeting many classes/prompts in parallel by minimizing an extremum objective (the negative minimum cosine similarity), and our maximally permitted adversarial perturbations are significantly higher than common for adversarial attacks since our proposed attack is intrinsically an off-manifold attack.

A similar objective in an adversarial context is minimized by Enevoldsen et al. (2023), who optimize the maximum logit of a neural classifier to generate adversarial examples for False Novelty and False Familiarity attacks in open-set recognition. Contrary to our work though, the aim of the optimization process is to increase or decrease the score of individual classes rather than many classes at once.

Bontrager et al. (2018) introduced the concept of latent variable evolution (LVE). The authors use the Covariance Matrix Adaption Evolution Strategy (CMA-ES) to perform stochastic search in the generator latent space of a Generative Adversarial Network (GAN) Goodfellow et al. (2014; 2020) to create *deep master prints*. Deep master prints are synthetic fingerprint images which match large numbers of real-world fingerprints, thus undermining the security of fingerprint scanners. Contrary to the approach of Bontrager et al. (2018), we use the decoder of a variational autoencoder (VAE) Kingma & Welling (2014) as well an extremum loss function to generate images from latents.

A number adversarial attacks by means of text patches and adversarial pixel perturbations have been performed on contrastively pre-trained multi-modal networks (Noever & Miller Noever, 2021; Daras & Dimakis,

2022; Li et al., 2021; Goh et al., 2021) Attacks on the text encoding were investigated by Daras & Dimakis (2022), where the authors show that one is able to generate images using nonsense-phrases in DALL-E 2. We believe this phenomena to be related to the issue of the modality gap between text and image embeddings, upon which our work builds. This modality gap in contrastively pre-trained multi-modal approaches has been documented originally by Liang et al. (2022), showing that the gap is caused by the inductive bias of the transformer architecture and reinforced by training a contrastive loss. While Liang et al. (2022) explicitly do not classify modality gaps as either beneficial or detrimental to a models performance, in our work we find that with respect to the vulnerability to off-manifold attacks, the modality gap should be mitigated. Nukrai et al. (2022) come to a similar conclusion upon finding that the modality gap causes instability when training a text decoder from CLIP embeddings.

Finally, in terms of the robustness of multi-modal neural networks Qiu et al. (2022) conducted an extensive evaluation of CLIP and CLIP-based text-to-image systems, where they come to the conclusion that CLIP and its derivatives are not robust with respect to distribution shifts.

## 3 Approach: CLIPMasterPrints

**Contrastive Language-Image Pre-Trained Models.** In production, a given model $C_\theta$, which has been trained using CLIP, is used to indicate how well a prompt or image caption $c$ describes the contents of an image $\mathbf{x}$ as follows. For each caption-image pair $(c, \mathbf{x})$, $C_\theta$ extracts a pair of corresponding vector embeddings $(\mathbf{f}(c), \mathbf{g}(\mathbf{x}))$ and computes their cosine similarity:

$$s(\mathbf{x}, c) = \mathcal{C}_\theta(\mathbf{x}, c) = \frac{\mathbf{g}(\mathbf{x})^\mathsf{T}}{\|\mathbf{g}(\mathbf{x})\|} \cdot \frac{\mathbf{f}(c)}{\|\mathbf{f}(c)\|}, \tag{1}$$

where a cosine similarity of 1 between $\mathbf{f}(c)$ and $\mathbf{g}(\mathbf{x})$ indicates an excellent match between prompt $c$ and image $\mathbf{x}$. On the other hand, $s(\mathbf{x}, c) \approx 0$ indicates that prompt and image are unrelated. In practice, it has been found though that $s(\mathbf{x}, c) = 1$ is hardly achieved, and even for well-fitting text-image pairs $s(\mathbf{x}, c) \approx 0.3$ (Schuhmann et al., 2021; Liang et al., 2022). The phenomenon of CLIP-trained models not being able to align matching text and image embeddings to achieve $s(\mathbf{x}, c) = 1$ has been studied extensively by (Liang et al., 2022), who refer to the underlying misalignment between image and text embedding vectors as the *modality gap* of multi-modal models.

**Exploiting the modality gap.** We exploit this modality gap, i.e. the misalignment of image and text embedding vectors, to mine fooling master images (CLIPMasterPrints) as follows.

In the latent space of $\mathcal{C}_\theta$, we aim to find an embedding $\mathbf{g}(\mathbf{x}_{\text{fool}})$ corresponding to a fooling master image $\mathbf{x}_{\text{fool}}$ for a number of matching text-image pairs $(c_1, \mathbf{x}_1), (c_2, \mathbf{x}_2), \ldots (c_n, \mathbf{x}_n)$ such that:

$$\frac{\mathbf{g}(\mathbf{x}_{\text{fool}})^\mathsf{T}}{\|\mathbf{g}(\mathbf{x}_{\text{fool}})\|} \cdot \frac{\mathbf{f}(c_k)}{\|\mathbf{f}(c_k)\|} > \frac{\mathbf{g}(\mathbf{x}_k)^\mathsf{T}}{\|\mathbf{g}(\mathbf{x}_k)\|} \cdot \frac{\mathbf{f}(c_k)}{\|\mathbf{f}(c_k)\|} \quad \text{for} \quad k \in [1, n].$$

The existence of a modality gap implies that there is a limit on how well the CLIP-trained model $\mathcal{C}_\theta$ can align $\mathbf{g}(\mathbf{x}_k)$, which is extracted from a vector on the image manifold $\mathbf{x}_k$ to $\mathbf{f}$, the models vector embedding of text prompt $c$ (Liang et al., 2022; Schuhmann et al., 2021).

We hypothesize that this apparent limit for vectors on the image manifold implies that if one were to search for vectors $\mathbf{x}_{\text{fool}}$ off manifold, one might find a vector that aligns better (and thus has a better cosine similarity score $s$) to all the captions $c_1, c_2, \ldots, c_n$, than any of the matching vectors on the image manifold $\mathbf{x}_1, \mathbf{x}_2, \ldots, \mathbf{x}_n$.

To test this hypothesis, we employ a number of different iterative optimization approaches for constructing $\mathbf{x}_{\text{fool}}$. In order to find an image that maximizes $s(\mathbf{x}_{\text{fool}}, c_k)$ for a set of $n$ different image captions $C = \{c_1, c_2, \ldots, c_n\}$ we minimize the loss function:

$$\mathcal{L}(\mathbf{x}) = -\min_{c_k \in C} s(\mathbf{x}, c_k). \tag{2}$$

To favor solutions where $\mathbf{x}$ matches all captions well, we use the min-operator over all $c_k$ rather than a sum or average. Our intention here is to avoid poor local minima, where $\mathbf{x}$ poses an excellent match for a small subset of captions and performs poor on the remaining ones.

**Stochastic gradient descent.** The most straight-forward approach to mine a fooling example $\mathbf{x}_{\text{fool}}$ is to minimize equation 2 by means of stochastic gradient descent (SGD) (and variants thereof). We start from a randomly initialized image $\mathbf{x}_{\text{fool}}^0$ and iteratively look for better fooling images by moving towards the direction of steepest descent on the loss surface. This direction is indicated by the subgradient of the loss function, which is defined as

$$\nabla_{\mathbf{x}}\left(-\min_{c_k \in C} s(\mathbf{x}, c_k)\right) = \nabla_{\mathbf{x}}(-s(\mathbf{x}, c_{\min})), \tag{3}$$

where

$$c_{\min} = \arg\min_{c_k \in C} s(\mathbf{x}, c_k). \tag{4}$$

This results in the iterative update rule

$$\mathbf{x}_{\text{fool}}^{t+1} = \mathbf{x}_{\text{fool}}^t - \eta \nabla_{\mathbf{x}_{\text{fool}}^t} \mathcal{L}, \tag{5}$$

where $\eta$ is the learning rate or step size.

While mining fooling examples using SGD variants is a proven and well-understood method (Nguyen et al., 2015), contrary to our approach, common approaches usually seek to increase or decrease the model's confidence w.r.t. a single particular class rather than targeting many classes at once.

**Latent Variable Evolution.** Attacking models using stochastic gradient descent bears the practical limitations of a whitebox-attack, i.e. the model's weights need to be known. As a complementary method, we also mine CLIPMasterPrints by means of a Latent Variable Evolution (LVE) approach (Bontrager et al., 2018; Volz et al., 2018). While the input dimensions of state-of-the-art neural networks are too large to be searched by a black-box evolutionary strategy (ES) on its own, in LVE, one searches the latent space of a generative model using ES. The latents found by the ES are then used to generate fooling example candidates, which are presented to the model under attack. From the model output, we compute the loss function in equation 2 and feed it back to the ES, which in turn creates new candidates. To evolve new solutions, we use the CMA-ES (Hansen & Ostermeier, 2001), a highly efficient and robust stochastic search method taking estimated second order information into account. We adapt the original LVE approach in two ways: First, by minimizing equation 2, we ensure that the mined image matches all targeted captions sufficiently well. Second, rather than using a custom-trained generative adversarial network (GAN) to generate fooling examples, we evolve our solution in the latent space of a pretrained variational autoencoder (VAE; Kingma & Welling, 2014). In more detail, we use decoder of StableDiffusion V1 (Rombach et al., 2022) to translate candidate latents into image space. Note that we do not apply any diffusion in this process, the VAE is in principle exchangeable with any other strong VAE. An overview of the approach is shown in Fig. 2, with Algorithm 1 in the Appendix detailing how to mine fooling examples with our LVE approach.

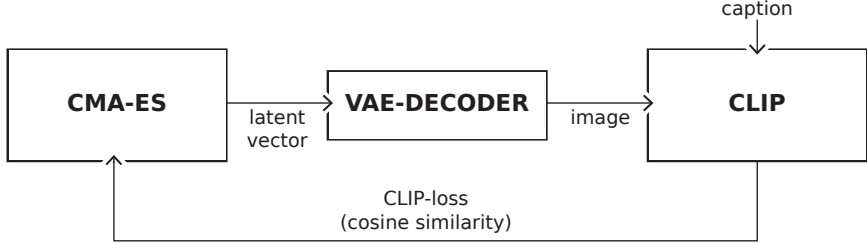

Figure 2: **CLIPMasterPrints Latent Variable Optimization.** CMA-ES is used to generate image candidates in the latent space of a pre-trained VAE. The generated latent vector is passed through the VAE's decoder and scored w.r.t. how well it fits to the caption using CLIP. The returned cosine similarity is thereafter fed back to CMA-ES.

**Projected gradient descent**. Finally, while CLIPMasterprints is essentially an off-manifold attack, we also evaluated projected gradient descent (PGD) (Kurakin et al., 2016; Madry et al., 2018) as a mining approach in order to investigate if it is also possible to mine fooling examples which are, to humans, much more similar to actual images. Contrary to SGD, we do not initialize $\mathbf{x}_{\text{fool}}^0$ randomly, but use an existing image $x_{\text{orig}}$ which we again update by moving towards the direction of steepest descent. However, as an additional step contrary to SGD, we attempt to keep our found solution close to $x_{\text{orig}}$. We do so by applying the PGD update rule

$$\mathbf{x}_{\text{fool}}^{t+1} = \Pi_{\mathbf{x}_{\text{orig}}+\epsilon}(\mathbf{x}_{\text{fool}}^t - \alpha\text{sign}(\nabla\mathcal{L}(\mathbf{x}_{\text{fool}}^t))), \tag{6}$$

to minimize equation 2, where the image $\mathbf{x}_{\text{fool}}^t \in [0, 255]^d$ is optimized in discrete representation, $\alpha$ is the discrete stepsize and $\epsilon$ is the size of the adversarial perturbation. $\Pi_{x+\epsilon}(a)$ is an operation used to keep $\mathbf{x}_{\text{fool}}$ close to $\mathbf{x}_{\text{orig}}$: It is defined as a element-wise clipping operation clipping each pixel $a_{i,j}$ of the input image $a$ into the range $[x_{\text{orig},i,j} - \epsilon, x_{\text{orig},i,j} + \epsilon]$ w.r.t the original image $x_{\text{orig}}$.

We permit for larger adversarial perturbation than commonly used in PGD attacks. We find that the approach does not work for too small adversarial perturbations, which again underlines the off-manifold-nature of the attack.

The CLIP models used in the experiments in this paper are pre-trained *ViT-L/14* and *ViT-L/14@336px* models (Radford et al., 2021).

## 4 Results

### 4.1 Experimental Setup

**Generating CLIPMasterPrints.** We test our approach to finding master images for both fooling CLIP on famous artworks and on ImageNet (Russakovsky et al., 2015) classes. For the artworks, we train a fooling master image to obtain a high matching score on the *ViT-L/14@336px* CLIP model (Radford et al., 2021) for 10 different text prompts, consisting of the titles of famous artworks and their authors. Famous artworks and their corresponding titles and artists were chosen for their familiarity: On the one hand, due to being widely known and therefore likely in the training data of the model, this approach ensures that CLIP scores between corresponding artwork-title pairs will be easily matched to each other, resulting in high cosine similarities obtained from the model for matching pairs. On the other hand, due to the uniqueness and distinctiveness of most images in both motive and style, it is unlikely that any two artworks will be confused by the model, resulting in low cosine similarities for image-text pairs that do not match.

We create one fooling master example for each mining approach introduced in Section 3. SGD is applied to a single randomly initialized image and optimized for 1000 iterations using Adam (Kingma & Ba, 2015) ($\beta_1 = 0.9$, $\beta_2 = 0.999$, $\epsilon = 10^{-8}$) at a learning rate of 0.1.

In our black-box approach, we search the latent space of the stable diffusion VAE (Rombach et al., 2022) for CLIPMasterPrints using CMA-ES for 18000 iterations. We flatten its 4 feature maps into a vector. Since images are encoded in this latent space with a downsampling factor of 8, our $336 \times 336$ images result in a $d = \frac{336}{8} \cdot \frac{336}{8} \cdot 4 = 7056$ dimensional search space. We initialize CMA-ES with a random vector sampled from a zero-mean unit-variance Gaussian distribution and choose $\sigma = 1$ as initial sampling variance. We follow the heuristic suggested by Hansen (2016) and sample $4 + 3 \cdot \log(d) = 4 + 3 \cdot \log(7056) \approx 31$ candidates per iteration.

Finally, for our PGD approach, we start from an existing image and again optimize for 1000 iterations using a stepsize of $\alpha = 1$ and a maximal adversarial perturbation of $\epsilon = 15$.

For generating fooling images for ImageNet classes, we mine a CLIPMasterPrint for 25, 50, 75 and 100 randomly selected ImageNet classes. To show that the approaches work independently of the chosen model weights and to speed up the more extensive experiments on ImageNet, the *ViT-L/14* model (Radford et al., 2021) was chosen, with a slightly smaller input pixel size of $224 \times 224$ pixels. For the SGD and PGD approaches, the parameters are identical as in the previous experiments. Our blackbox-LVE approach mines for 50,000 iterations. The remaining parameters are the same as in the previous experiment, except since

smaller images with a resolution $224 \times 224$ pixels were generated, the corresponding search space consists of $\frac{224}{8} \cdot \frac{224}{8} \cdot 4 = 3136$ dimensions. This yields a population size of $4 + 3 \cdot \log(3136) \approx 28$ candidates per iteration.

**Mitigation by bridging the modality gap.** As we hypothesise a model's vulnerability to be connected to it's modality gap, as a mitigation approach, we attempt to bridge the used *ViT-L/14* model's gap by shifting the centroids of image and text embeddings as suggested in Liang et al. (2022). In more detail, Liang et al. (2022) decrease the gap between image and text vectors by moving them toward each other along a so-called gap vector

$$\Delta_{\mathrm{gap}} = \bar{\mathbf{f}} - \bar{\mathbf{g}} \ , \tag{7}$$

where $\bar{\mathbf{f}}$ and $\bar{\mathbf{g}}$ are the centroids of image and text embeddings, respectively. We extract $\bar{\mathbf{f}}$ and $\bar{\mathbf{g}}$ for the ImageNet training data and labels. We attempt to bridge the model's modality gap by computing

$$\mathbf{x_i}' = \mathbf{x_i} - \lambda\Delta_{\mathrm{gap}} \tag{8}$$

and

$$\mathbf{y_i}' = \mathbf{y_i} + \lambda\Delta_{\mathrm{gap}}, \tag{9}$$

as shifted image and text embeddings, respectively. $\lambda = 0.25$ is a hyperparameter chosen such that the model retains its original accuracy as much as possible while bridging the gap.

**Mitigation by sanitizing model inputs.** Apart from increasing the robustness of a CLIP-trained model itself, a further possible route to mitigate adversarial attacks could be to sanitize the model's inputs by automatically detecting adversarial examples and sorting them out early. We build a custom training set from ImageNet subsets and train a ConvNet to detect the visible artifacts of PGD-mined adverserial images. In more detail, we create train validation and test sets of 60000, 10000 and 10000 images respectively, each from a subset of the ImageNet train set. We do so by using each image of the respective subsets to initialize the PGD adversarial mining process for 25 randomly selected ImageNet target classes. To keep the required amount of compute feasible, given the large amount of CLIPMasterPrints to be mined, we only mine for 100 iterations per image and target a smaller CLIP model than used in the remainder of this work, i.e. *ViT-B/32*. In the finished dataset, in all subsets (train, validation and test) 50% of all images are mined CLIPMasterPrints, while remaining images are the templates used to initialize the mining process, i.e. randomly chosen images from the Imagenet train and validation sets. As a classifier, we use a Imagenet-pretrained VGG19 ConvNet (Simonyan & Zisserman, 2015) with batch normalization (Ioffe & Szegedy, 2015) between each convolutional layer and activation function. We refine the model for 1 epoch using Adam at a learning rate of $10^{-3}$ and a batch size of 152. We use default momentum parameters $\beta_1 = 0.9$, $\beta_2 = 0.999$, $\epsilon = 10^{-8}$ for Adam and do not apply any L1 or L2 regularization to the weights.

In addition to the two introduced approaches above, a third, somewhat less effective mitigation approach is discussed in the appendix in Section A.2

### 4.2 Performance of CLIPMasterPrints

Fig. 1 shows the cosine similarities between titles and artists of famous artwork and the actual artwork as well as a baseline image and our generated CLIPMasterPrints (denoted by red frames). All artworks are assigned their correct titles by the CLIP model: artworks and their respective titles exhibit a significantly higher cosine similarity (of about 0.3) than randomly paired titles and paintings. Our noise baseline exhibits scores between 0.13 and 0.18 for all title-captions, but interestingly at times shows higher scores compared to artworks with mismatched captions. All mined CLIPMasterPrints yield cosine similarities $> 0.33$ and consequentially outperform the original artwork for each title-caption. Yet, we find large differences in-between the performance of samples mined with different approaches.

The fooling image mined through SGD (Fig. 3d) performs best, followed by PGD, which, despite superficial unnatural patterns, clearly resembles a natural image more closely (Fig. 3e). LVE performs least well, while requiring a significantly higher number of iterations. However, it still outperforms the original artworks. An explanation can be found in the more constrained optimization space of the VAE latents as well as the absence of gradient information. All three fooling master examples achieve a higher score than all actual

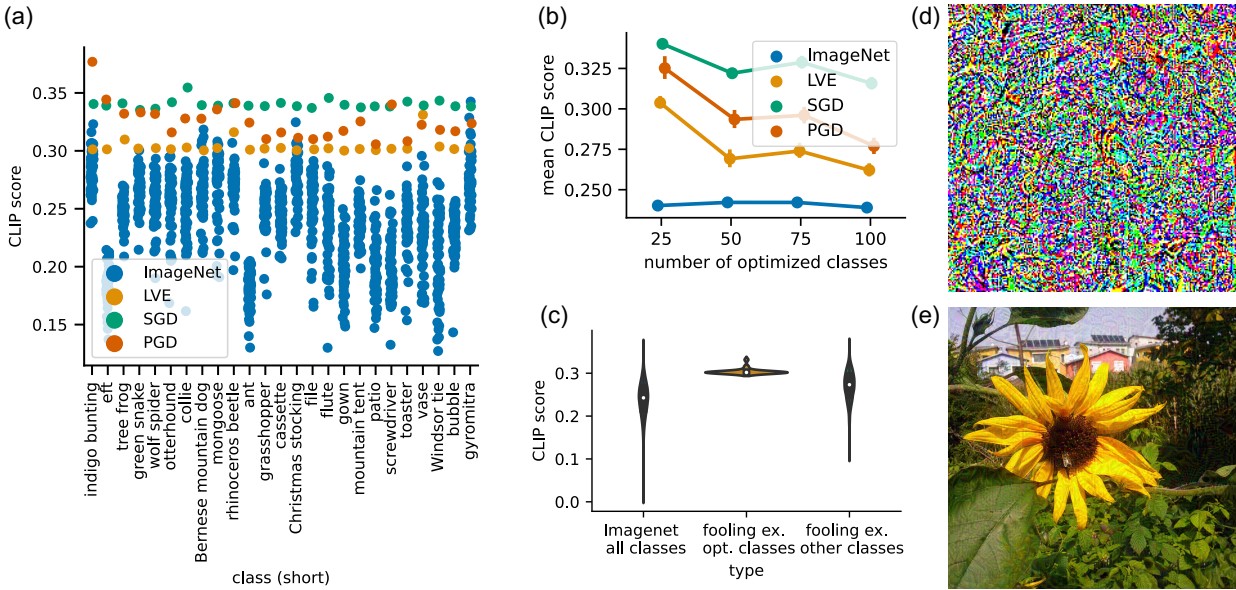

Figure 3: (**a**) Cosine similarity of three trained fooling images for 25 targeted classes using SGD, LVE and PGD approaches respectively, as well as similarities for ImageNet validation set images of the same classes. With a few exceptions, each CLIPMasterPrint fooling image outperforms all images in terms of CLIP score for the targeted text labels. *Note that the same fooling image is used for all class label categories.* (**b**) Average cosine similarity between ImageNet class captions and fooling image as a function of the number of classes considered during optimization for SGD, LVE and PGD methods. Average similarity score between captions and images in the ImageNet validation set labelled with targeted class labels for comparison. Score remains stable up to 75 targeted classes, after which it gracefully declines. Due to CLIPMasterPrints generalizing to semantically related labels, the achieved average score remains robust, even if more related labels are added. (**c**) Generalization of LVE-mined image targeting 25 ImageNet classes. The mined CLIPMasterPrint achieves high CLIP scores even for ImageNet class labels which have not been explicitly targeted, as shown by score distributions of matched label-text pairs in the ImageNet validation set and score distributions between CLIPMasterPrint and untargeted ImageNet labels being almost identical. Examples of unrecognizable (**d**) and recognizable images (**e**) created by SGD and PGD, respectively.

artworks and would be chosen over these images when prompting the model to identify any of the targeted artworks next to the fooling examples.

Our results for ImageNet labels are similar. Fig. 3a shows the CLIP-returned cosine similarities of the fooling master image trained on 25 ImageNet labels as a point plot for both gradient-based (SGD, PGD) and blackbox (LVE) approaches. The cosine similarities of the images of the respective labels found in the ImageNet validation set have been added for reference. For almost all classes, the two images mined with SGD and PGD outperform the entirety of the images within the respective class in terms of the similarity score. The black-box LVEimage on the other hand, while performing somewhat worse, still outperforms the entirety of images for most classes.

As a performance measure over all optimized classes, we compute the percentage of outperformed images (POI, i.e. the percentage of images in targeted classes in the validation set with a lower CLIP score than the fooling image) for all three fooling images. We find that our SGD and PGD images exhibit an accuracy of 99.92% and 99.76%, respectively, while the LVE images achieve an accuracy of 97.92% which is in line with our observations from Fig. 3a.

These results demonstrate that CLIP models *ViT-L14* and *ViT-L14@336px* can be successfully fooled on a wider range of classes using only a single image.

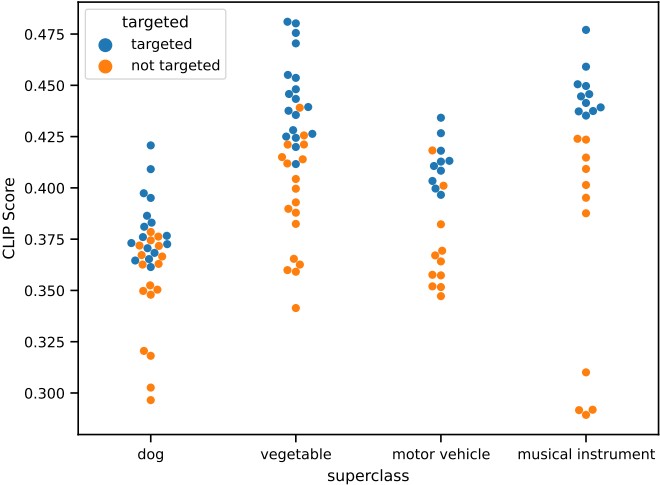

Figure 4: CLIPMasterPrints targeting related subterms of the four nouns "dog", "vegetable","motor vehicle" and "musical instrument". The four mined images achieve only slightly lower scores on most untargeted subterms compared to targeted subterms. CLIPMasterPrints therefore generalize to untargeted, but semantically related prompts.

### 4.3 Generalization to semantically related prompts and labels

To investigate whether the mined images also generalize to semantically related classes that were not directly considered in the optimization process, we also visualized the estimated distributions of CLIP similarity scores per class for both targeted and untargeted classes (Figure 3c). While the distribution of cosine similarities over all true classes in the ImageNet validation set (i.e. the cosine similarities for all *ground-truth matched* image-text pairs in the validation set) is long-tailed, the distribution for scores of the CLIPMasterPrint (mined with LVE) for targeted classes is confined to a small interval around 0.30, which is also the score achieved on targeted labels as seen in Fig. 3a. Considering the distribution of scores for the fooling image on all 975 not targeted classes, we see that while the distribution is long-tailed as well, most values seem to be confined to the range between 0.2 and 0.3, with a mean around approx 0.27. The strong similarity between the distribution of ImageNet image-text pairs and the cosine similarity of our mined CLIPMasterPrint on untargeted classes indicate a strong generalization effect. Computing the POI for the 975 untargeted classes in the ImageNet validation paints a similar picture. We find that our SGD, LVE and PGD mined fooling images score-wise outperform 87.3, 74.02 and 88.63% of images averaged over the 975 classes, respectively. In summary, the fooling images achieve moderate to high scores on untargeted class labels. A potential explanation is due to the classes of the ImageNet dataset being derived as a subset from tree-like structures in WordNet (Miller, 1995), CLIPMasterPrints generalize on many of these classes due to them being semantically related to their targeted labels.

We investigate this hypothesis by training five additional CLIPMasterPrints using PGD on WordNet hyponyms (related subterms) of the four nouns "dog", "vegetable", "motor vehicle" and "musical instrument". We train one CLIPMasterPrint for half of the hyponyms of a particular noun and then evaluate it's performance on the remaining ones. Figure 4 shows the results.

As one can see, apart from a few exceptions, untargeted subterms only obtain slightly lower or at times even on par scores than than the targeted classes. We therefore conclude that mined CLIPMasterPrints can indeed target semantically related, not directly optimized classes as well.

### 4.4 Performance as number of targeted prompts increases

Our results demonstrate that CLIP models are vulnerable to fooling master images, and that fooling effects generalize to semantically related labels or nouns. We thus investigate how the average cosine similarity on targeted classes deteriorates, as the number of targeted class labels increases. Fig. 3b shows the average CLIP score targeting 50, 75, and 100 randomly sampled ImageNet classes versus the total number of targeted labels for all evaluated approaches. For all approaches, the average score exhibits an initial decrease around

50 classes after which it slightly rises for 75 and then slightly decreases for 100 classes again. An explanation for this observation can be found in the generalization effects observed above: assuming that subsets of the targeted labels or prompts are sufficiently semantically related, due to the generalization of the fooling example, the achieved average score remains robust, even if more related labels are added. For more results, see also Section A.3 in the appendix, where we find that CLIPMasterPrints can be mined to target hundreds of ImageNet classes.

## 4.5 Mitigation

**Bridging the modality gap.** We find shifting centroids of image and text embeddings along a computed gap vector as discussed above (Eq. 7, 8 and 9), to be an effective countermeasure against CLIPMasterPrints while preserving CLIP performance. Table 1 shows the percentage of outperformed images (POI) for CLIP-

Table 1: Pct. of outperformed images for different optimization approaches on the validation set.

| Method | POI, $\lambda = 0$ | POI, $\lambda = 0.25$ |
|---|---|---|
| SGD | 99.92% | 3.2% |
| LVE | 97.92% | 1.28% |
| PGD | 99.76% | 1.92% |
| SGD, $\lambda = 0.25$ | 76.64% | 63.2% |
| LVE, $\lambda = 0.25$ | 48.56% | 38.64% |
| PGD, $\lambda = 0.25$ | 52.88% | 44.64% |

MasterPrints mined both with and without shifting embeddings in the model. Not only fooling examples mined on the regular model (Rows 1, 2 and 3 for SGD, LVE and PGD respectively) do not work anymore on the model with shifted embeddings (the POI drops dramatically), but also newly mined examples from a model with shifted embeddings (Rows 4, 5 and 6) show a significant drop of roughly 35 to 55 percentage points in POI. Shifting embeddings therefore can be considered an effective mitigation strategy. When considering the scores of the different images mined on the shifted model, we find the the SGD image performs best, followed by the PGD image, with the LVE approach performing least well. One may expect the PGD image to perform best under a mitigated modality gap, since it is closest to a natural image. Yet, when we compute the cosine similarity between the latent of the original image $\mathbf{x}_{\text{orig}}$ and the mined image $\mathbf{x}_{\text{PGD}}$, we find that

$$\frac{\mathbf{g}(\mathbf{x}_{\text{orig}})^{\mathsf{T}}}{\|\mathbf{g}(\mathbf{x}_{\text{orig}})\|} \cdot \frac{\mathbf{g}(\mathbf{x}_{\text{PGD}})}{\|\mathbf{g}(\mathbf{x}_{\text{PGD}})\|} = 0.29.$$

Despite its similarity to $\mathbf{x}_{\text{orig}}$, the mined adversarial image therefore is not located on the image manifold in the models latent space. Furthermore, we observe that when mining CLIPMasterPrints by means of PGD, using adversarial perturbations $\epsilon <= 10$ pixels, i.e. producing solutions closer to the original image, yields poor results. We consider this a further indicator that CLIPMasterPrints need to be located off the models latent image manifold.

In summary, we argue that the fact that the mitigation technique we use here has originally not been proposed as a defense against adversarial examples, but was rather used to investigate modality gaps in general Liang et al. (2022), adds strong support our original hypothesis that the vulnerability of a CLIP model to CLIPMasterPrints is closely related to the modality gap.

**Sanitizing model inputs.** Apart from increasing the robustness of a CLIP-trained model itself, a further possible route to mitigate adversarial attacks is to sanitize the model's inputs by automatically detecting adversarial examples and sorting them out early. We build a custom training set from ImageNet subsets and train a classifier to detect the visible artifacts of PGD-mined adverserial images (for details on the setup see Section 4.1). We find that we are able to detect whether an image is a PGD-mined CLIPMasterPrint or a "harmless" image with 99.01% accuracy on the test set, which makes the proposed approach an effective mitigation strategy to sanitize inputs of real-world systems.

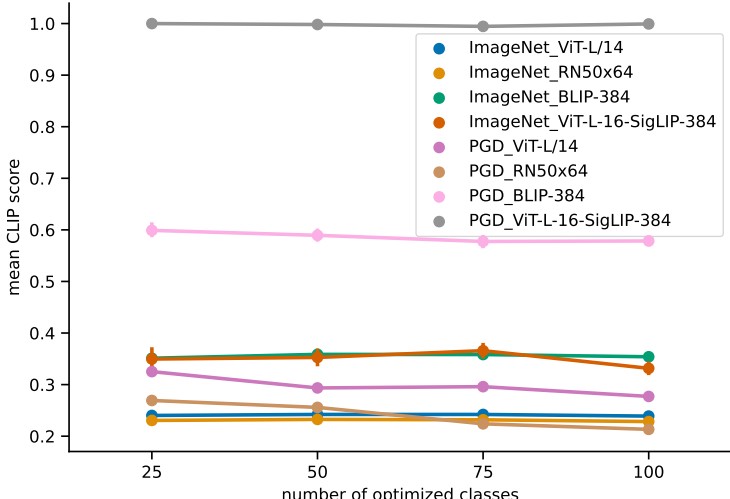

Figure 5: Performance of CLIPMasterPrints mined using PGD on *CLIP-RN50x64*, *BLIP-384* and *ViT-L-16-SigLIP-384* for 25, 50, 75 and 100 ImageNet classes respectively, *CLIP-ViT-L/14* and Imagenet baselines for comparison. Models that use ResNet rather than visual transformers as well as newer models improving upon CLIP are nevertheless vulnerable to CLIPMasterPrints.

### 4.6 Attacking different architectures and training approaches

To demonstrate that CLIPMasterPrints are not an isolated phenomenon limited to the investigated architecture or training approach, we mine additional CLIPMasterPrints on a further CLIP-trained model using an ensemble of 64 ResNet50 networks to as an image encoder (*CLIP-RN50x64*). Furthermore we do the same for models trained on recently proposed improvements or CLIP, namely BLIP Li et al. (2022) and SigLIP Zhai et al. (2023). Figure 5 shows the results. All evaluated models remain vulnerable to CLIPMasterPrints: The CLIP-model using ResNet image encoding (*CLIP-RN50x64*) seems to be somewhat less vulnerable than the transformer-based CLIP *ViT-L/14*, but is still on par with the ImageNet baseline. For both newer approaches, *BLIP-384* and *ViT-L-16-SigLIP-384*, the PGD-mined CLIPMasterPrints outperform the ImageNet baselines significantly. These results clearly show that vulnerability CLIPMasterPrints is not just limited to CLIP, but also concerns more recently proposed models trained with related contrastive approaches.

## 5 Potential Attack Scenarios

As emphasised by Radford et al. (2021), next to zero-shot-prediction, a highly relevant application of CLIP is zero-shot image retrieval, which offers plenty of attack surface by means of CLIPMasterPrints. In more detail, inserting a single CLIPMasterPrint into an existing database of images could potentially disrupt the system's functionality for a wide range of search terms, as for each targeted search term the inserted fooling master image is likely to be the top result. When inserting several CLIPMasterPrints into the database, even the top n results could consist entirely of these adversarial images rather than the true results. While this is also possible when inserting "regular" adversarial examples, the amount of examples needed for an attack using CLIPMasterPrints is orders of magnitude lower than for regular adversarial examples. Practical malicious applications of this vulnerability could be 1) censorship of images related to a list of censored topics, 2) adversarial product placement: targeting a variety of searched brands to advertise a different product as the top result, or 3) disruption of service: introducing a larger number of unrecognizable CLIPMasterPrints for a wide range of topics, resulting in unintelligible results for many queries, reducing the quality of service of an image retrieval system.

A further interesting issue can be raised with respect to whether fooling examples, which can be recognized by the user as such, i.e. they do not resemble natural images, or show unnatural artifacts which makes them recognizeable as adversarial examples to an attentive user, pose a real-world threat to AI systems in

production. Here we argue that even if fooling images can be recognized by humans, there still remain many ways for an attacker to introduce adversarial examples where no human supervision or control is present. For instance, introducing CLIPMasterPrints into a database could be as simple as putting images online to be crawled by search engines or uploading them through webforms. Impairing the function of the attacked system, or censoring particular images in the system can in this case still be achieved using unrecognizeable fooling images. We show in Section 4.5, that sanitizing the model inputs by training an additional classifier to detect CLIPMasterPrints can be an effective way to mitigate threat surface. Of course this approach bears the drawback that an additional classifier needs to be trained and deployed in production, under which perspective further investigation into increasing the robustness of CLIP-trained models is desireable.

In cases where human supervision is present on the other hand, adversarial examples with slight artifacts may be spotted by an attentive user, but might still fool a distracted or technologically less proficient user. Furthermore, a cunning attacker might choose a template image where resulting artifacts are difficult to make out. We invite the reader to consider the PGD-mined CLIPMasterPrints in Figure 6e and 6f in the appendix and form their own opinion on how prominent the resulting artifacts are, and whether they could be missed by an inattentive user or hidden by selecting an appropriate image.

# 6  Discussion and Future Work

This paper demonstrated that CLIP models can be successfully fooled on a wide range of diverse captions by mining fooling master examples. Images mined through both gradient-based (SGD, PGD) as well as gradient-free approaches (LVE) result in high confidence CLIP scores for a significant number of diverse prompts, image captions or labels. While the gradient-free approach performed slightly worse, it does not require access to gradient information and therefore allows for black-box attacks.

We found that the modality gap in contrastively pre-trained multimodal networks (i.e. image and text embeddings can only be aligned to a certain degree in CLIP latent space) plays a central role with respect to a model's vulnerability to the introduced attack. Low cosine similarity scores assigned to well-matching text-image pairs by a vulnerable model imply that off-manifold images, which align better with a larger number of text embeddings, can be found. PGD-mined images, while being appearing meaningful to humans, are nevertheless found to be off the latent image manifold of the attacked model. The off-manifold nature of the attack is also supported by the observation that information in fooling examples is distributed throughout the whole image for all targeted prompts, rather than locally at different places for each prompt (see Section A.4 in the Appendix), making the mined images vulnerable to occlusion and cropping.

We show that a possible way to exploit the off-manifold nature of the attack for possible mitigation approaches is to train a classifier to detect the artifacts introduced by the adversarial mining process in order to sanitize model inputs. This way we are able to reliably distinguish CLIPMasterPrints from "harmless" images with an accuracy 99.01% for our test set. While this approach could be highly effective for real-world systems, it bears the drawback that an additional classifier needs to be trained and deployed in production. Under that perspective further investigation into increasing the robustness of CLIP-trained models is desireable.

In terms of increased model robustness, our results demonstrate that the effects of CLIPMasterPrints on the model can be mitigated by closing the gap between centroids of image and text embeddings respectively. While Liang et al. (2022) do not explicitly classify modality gaps as either beneficial or detrimental to a models performance, our results support the hypothesis that the modality gap leaves CLIP models vulnerable towards CLIPMasterPrints. Thus efforts to mitigate modality gaps even further, while preserving model performance, is a critical future research direction.

Finally, our mined CLIPMasterPrints seem to not only affect the prompts they target, but also generalize to semantically related prompts. In combination with the observation that recent improvements to CLIP such as BLIP and SigLIP are vulnerable as well, this generalization effect additionally increases the impact of the introduced attack. In conclusion, further research on effective mitigation strategies as well as the vulnerability of additional related models is needed.

## Reproducibility Statement

We supply our code with instructions on how to reproduce our experiments as supplementary material. The code is also available at https://github.com/matfrei/CLIPMasterPrints.

## Broader Impact Statement

The approaches introduced in this paper could be used to mount attacks that misdirect CLIP models in production. For instance, an attacker could manipulate the rankings of a CLIP-based image retrieval system resulting in injected CLIPMasterPrints being the top result for a wide range of search terms. This could be exploited in malicious ways for censorship, adversarial marketing and disrupting the quality of service of image retrieval systems (for details see Section 5). Nevertheless, we argue that publishing this work is a necessary step towards understanding the risks of using CLIP-trained models in real-world applications. We also propose and evaluate mitigation strategies and hope that our work will inspire others to build on those to make them even more effective in the future.

## Acknowledgments

This work was supported by a research grant (40575) from VILLUM FONDEN.

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

## A  Appendix

### A.1  Mined fooling images

Figure 6 shows a selection of mined fooling images in good quality.

### A.2  Other mitigation approaches

As an additional mitigation approach, we explored making the *ViT-L/14* model robust by adding fooling images to the train set.

**Experiment setup.** First, we refine the model on the ImageNet train set, where we add for every batch presented to the network, both a random noise image as well as an LVE fooling example. Both the noise image and the fooling example get labeled with a special *<off-manifold>*-token in order to have the model bind off-manifold inputs to that token rather than any valid ImageNet label. At every forward step of the model, we generate a new random noise image by feeding zero-mean unit-variance Gaussian Noise into the decoder part of our generating autoencoder. The fooling example on the other hand is generated by running CMA-ES in the loop with the training process. We start out with the best-found previous solution and run one iteration of CMA-ES for every forward step to update the fooling example to the changed training weights of the model. This setup creates a similar optimization process as found in GANs where both models

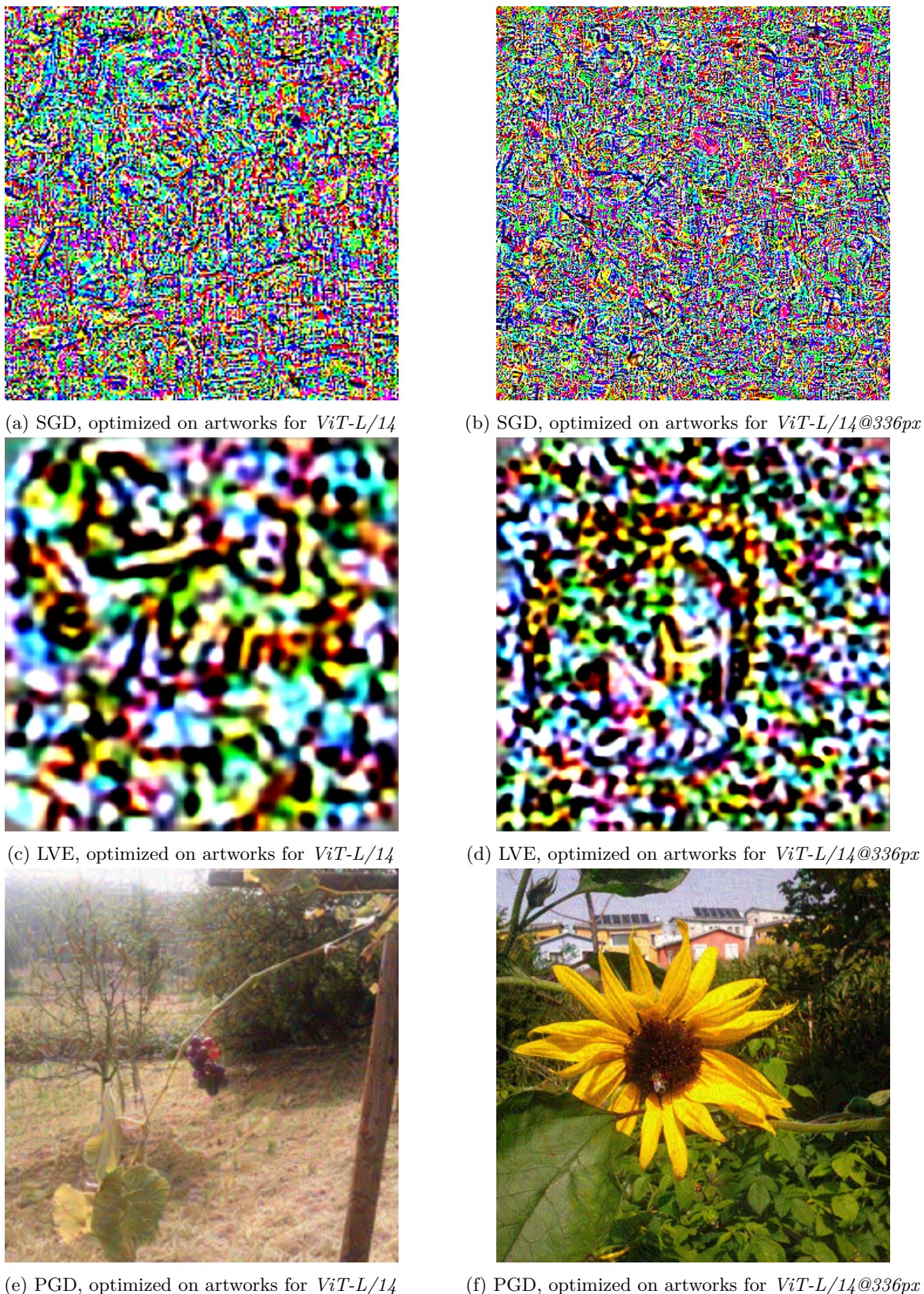

(a) SGD, optimized on artworks for *ViT-L/14*      (b) SGD, optimized on artworks for *ViT-L/14@336px*

(c) LVE, optimized on artworks for *ViT-L/14*      (d) LVE, optimized on artworks for *ViT-L/14@336px*

(e) PGD, optimized on artworks for *ViT-L/14*      (f) PGD, optimized on artworks for *ViT-L/14@336px*

Figure 6: Examples of CLIPMasterPrint images mined through SGD (a,b), LVE (c, d) and PGD (e, f). The complementary approaches are able to produce fooling images unrecognizable to humans (a–d) and images that resemble natural images but that display some artefacts perceptible to human eyes (e, f).

attempt to outperform each other. We refine the model for 1 epoch using Adam at a learning rate of $10^{-7}$ and a batch size of 20. We regularize the model with a weight decay of $\gamma = 0.2$ and set Adam momentum parameters as described in (Radford et al., 2021): $\beta_1 = 0.9$, $\beta_2 = 0.98$, $\epsilon = 10^{-6}$. Furthermore, we utilize mixed-precision training Micikevicius et al. (2018). Hyperparameters for CMA-ES are identical to the ones used to mine the original fooling image. Finally, after refining the model, we mine a new fooling example from scratch for the updated model. We do so to test the model's robustness not only to the original fooling images, but fooling images in general.

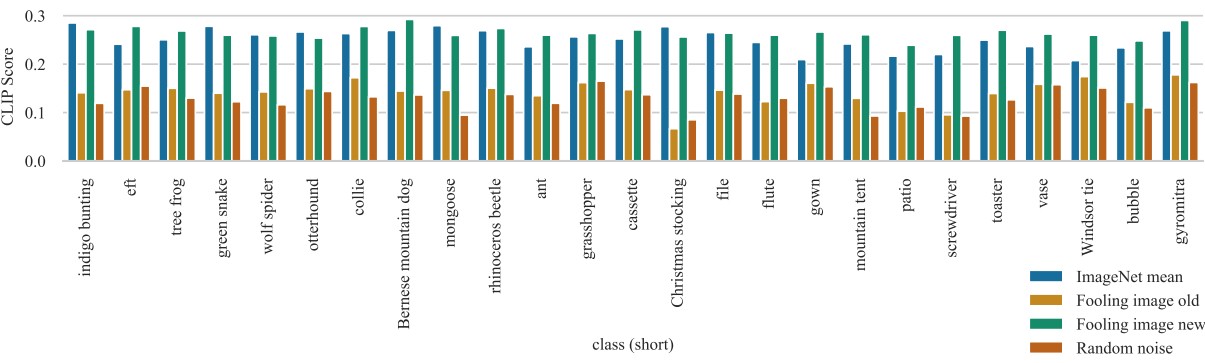

Figure 7: CLIP scores for fooling examples mined before and after refinement with off-manifold token. While mapping existing fooling examples to special tokens can mitigate their impact, the model is still vulnerable to new fooling images

**Results.** Fig. 7 shows the CLIP scores of our refined model, which has been trained to align off-manifold vectors to a special token, in order to mitigate the model's vulnerability to fooling master examples.

Shown are the average CLIP score on the ImageNet validation set, the CLIP score for the original fooling example, the score for a fooling example trained after refinement, as well as the score of a random noise image for each targeted ImageNet label respectively. Due to the newly introduced *<off-manifold>*-token, both noise and the original fooling examples are suppressed by the model and score significantly lower as the mean label score on the ImageNet validation set.

The newly mined fooling example on the other hand has not been suppressed at all by the refined model and exhibits scores similar to the ImageNet mean for all labels. The results suggest that our mitigation strategy is sufficient to mitigate existing fooling examples, yet fails to be effective as new fooling examples are mined from the updated model.

### A.3 Targeting up to 1000 classes

In order to explore the behavior for CLIPMasterPrints when going beyond 100 classes, we mined further fooling images targeting 25, 50, 75, 100, 250, 500, 750 and 1000 classes using PGD. To account for variations in performance, we mine 10 CLIPMasterPrints for each number of target classes, where we randomly vary both fooling image initialization as well as the permutation (i.e. order) of the target classes (random seeds used: $0-9$). To avoid the mining algorithm to overstep, we save image with the smallest loss as in equation 2 after 1000 iterations. The violin plot Figure 8 shows distributions of the entirety of obtained CLIP scores for all fooling examples, as a function of the number of classes target. For comparison, the distributions of all scores obtained by paired images of the entirety of targeted classes in the Imagenet validation set has been added. While, as expected, the overall performance of mined CLIPMasterPrints declines as more classes are targeted, even the image targeting 1000 classes manages to achieve similar or better scores than the majority of the images in the ImageNet training set. We can therefore conclude that, when targeting hundreds of classes, CLIPMasterPrints do not necessarily outperform any and all images in these classes, they still score significantly higher than a large portion of images in question.

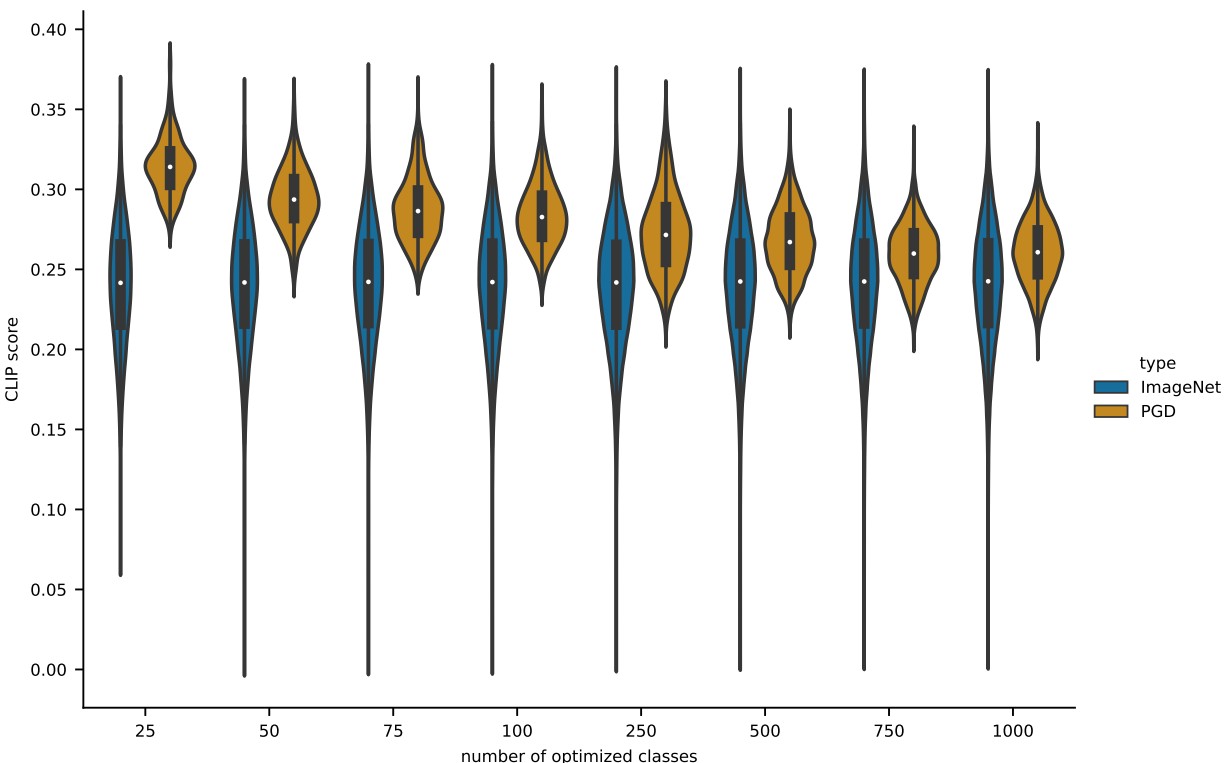

Figure 8: Distribution of scores for CLIPMasterPrints mined using PGD for 25, 50, 75, 100, 250, 500, 750 and 1000 ImageNet classes respectively, distribution scores for Imagenet images in the targeted classes for comparison. While the score declines as the number of targeted classes increases, even CLIPMasterPrints targeting all 1000 Imagenet classes achieves largely on par or better scores than the majority of images in the ImageNet training set.

## A.4    Analysis of information distribution in CLIPMasterPrints

To understand how information is distributed in the found fooling master examples, we create occlusion maps (Zeiler & Fergus, 2014; Selvaraju et al., 2017) of the fooling master example trained on the titles of famous artworks (Fig. 9). As we blur $75 \times 75$ rectangles of the fooling master image in a sliding-window-manner with a 2 pixel stride and a large ($\sigma = 75$) Gaussian blur kernel, we measure the change in cosine similarity as returned by the *ViT-L14@336px* model. As a reference, the same procedure is performed on a number of artworks the fooling image is intended to mimic. Blurring any part of CLIPMasterPrint results in a significant decrease (between 0.1 and 0.2) of the resulting similarity score of the model. It os noteworthy that the image optimized u sing LVE seems to be more robust to occlusions than images obtained by SGD and PGD methods. This can be explained by the more grainy and contrasted patterns in the LVE image. The individual increases and decreases for actual artworks on the other hand are more moderate and vary based on the location in the image.

For the *Random noise image* prompt, which has been excluded from optimization, blurring parts of the image results in significantly smaller changes in model output score. Interestingly, the mined CLIPMasterPrints react differently to occlusion based on the used optimization approach. For the SGD image, blurring different regions of the image affects decreases the score in some regions, while it increases it in others. As the SGD image closely resembles a pure noise image, it seems intuitive that blurring certain parts of the image decreases model similarity, yet it is unclear why blurring parts of the image increases the similarity. For the LVE image on the other hand, blurring does not result in improving scores, but again, different regions

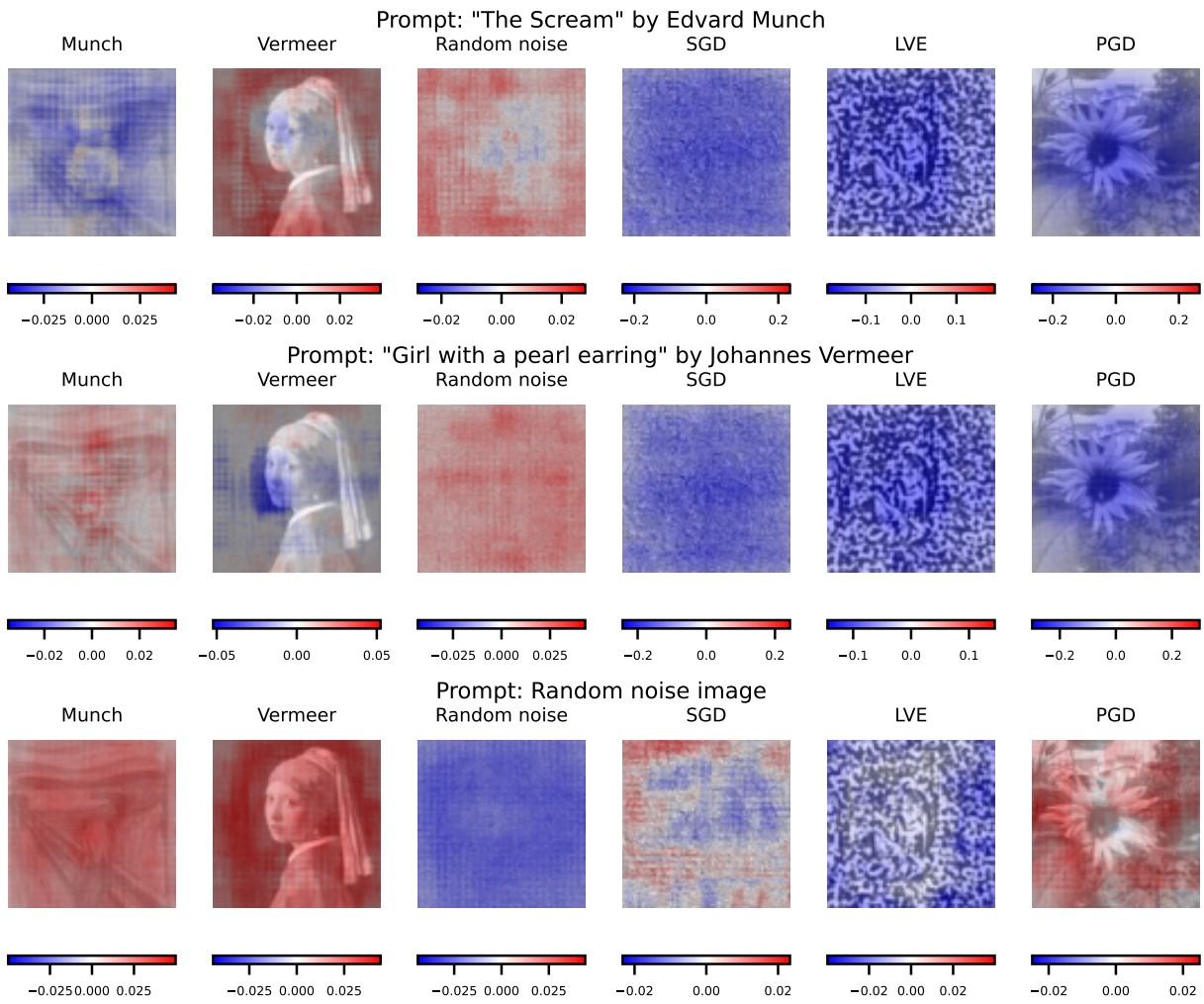

Figure 9: Occlusion maps for famous artworks, random noise baseline, and mined CLIPMasterPrints for different prompts. Note that, while each row shows the same CLIPMasterPrint, occlusion maps vary for different prompts. Increases in cosine similarities when blurring out a certain part of the image are denoted in red, decreases are shown in blue. Information in the CLIPMasterPrint is distributed over the whole image; no individual regions in the image that can be mapped to a particular prompt.

respond differently to the noise prompt. Finally, for large parts of the PGD image, the similarity score improves as information in the image is erased through blurring.

While these results demonstrate that the way that CLIP assigns similarity is often far from intuitive to humans, we can conclude that information from CLIPMasterPrints resulting in high CLIP confidence scores is spread throughout the image for all captions, and is quite sensitive to occlusions and cropping.

While CLIP has learned to deal with blurring and occlusions in natural images due to the large amount of sufficiently varying images presented during training, this robustness does not translate to other patterns such as noise and adversarial images. We have shown above that blurring the latter two results in a significant misalignment of the resulting vector in relation to the text vectors it has been targeting.

### A.5 Pseudocode for black-box mining of CLIPMasterPrints

Algorithm 1 illustrates our black-box approach to mining CLIPMasterPrints as pseudocode listing.

---

**Algorithm 1** Black-box approach to find CLIPMasterPrints

---

**Input:** initial vector $\mathbf{h}_0 \sim \mathcal{N}(0,1)$,
list of objective prompts $c_1, c_2, \ldots, c_n$,
number of to-be-run iterations $i_{max}$
pre-trained CLIP model $\mathcal{C}_{\theta_1}$
pre-trained image decoder $\mathcal{D}_{\theta_2}$
Initialize CMA-ES with $\mathbf{h}_0$
**for** i = 1 **to** $i_{\max}$ **do**
  Generate candidates $\mathbf{h}_1, \mathbf{h}_2, \ldots, \mathbf{h}_n$ using CMA-ES mutation
  Decode images $\mathbf{x}_1, \mathbf{x}_2, \ldots, \mathbf{x}_n$ from $\mathbf{h}_1, \mathbf{h}_2, \ldots, \mathbf{h}_n$ using $\mathcal{D}_{\theta_2}$
  **for all** $\mathbf{x}_j$ **in** $\mathbf{x}_1, \mathbf{x}_2, \ldots, \mathbf{x}_n$ **do**
    Set $s_{j,\min} = \infty$
    **for all** $c_k$ **in** $c_1, c_2, \ldots, c_n$ **do**
      Set $s_{j,k} = \mathcal{C}_{\theta_1}(\mathbf{x}_j, c_k)$
      **if** $s_{j,k} < s_{j,\min}$ **then**
        Set $s_{j,\min} = s_{j,k}$
      **end if**
    **end for**
  **end for**
  Update CMA-ES statistics with $s_{1,\min}, s_{2,\min}, \ldots, s_{n,\min}$
**end for**

---

