# OpenReview forum: "Fooling Contrastive Language-Image Pre-Trained Models with CLIPMasterPrints"
_TMLR — Accepted by TMLR_

### Review · Reviewer_FGxL · 2024-01-18

**Summary Of Contributions:**

This work follows the usual template of the many other works in adversarial ML: propose new attacks, study defenses, draw a possible connection to real-world systems. This work empirically studies adversarial attacks and mitigations on CLIP models. The proposed attacks involve finding a "fooling image" (or CLIPMasterPrint), an image that achieves high confidence scores across several classes (prompts) despite not resembling any of said classes/prompts.
The optimization proposed to find such fooling images is the minimum cosine similarly between the embedded image and embedded prompts over a collection of given prompts. Three methods of solving the optimization problem are studied: SGD, PGD, and an evolutionary strategy in latent space called LVE.

An experiment on the generalization of fooling images to classes/prompts unseen at attack time is made, finding that fooling images can sometimes transfer to "semantically related" classes. An investigation of a mitigation strategy based on the "modality gap" is made, where it is found that reducing this gap leads to less effective attacks. The potential practical relevance of such attacks is envisioned to involve manipulation of image retrieval systems and others.

**Audience:**

Yes

**Broader Impact Concerns:**

A Broader Impact Statement is given. No further concerns.

**Claims And Evidence:**

Yes

**Requested Changes:**

**Critical**

Can the proposed fooling images be reliably detected? If so, does this not eliminate the envisioned risk to real-world systems?

Revise the experiment in Section 4.2 to explicitly test whether generalization to "semantically related" or not classes using the Wordnet hierarchy.

**Strengthen**

Fix legend in Figure 4.

Please clean up writing as described above.

**Strengths And Weaknesses:**

**Strengths**

Studying attacks/defenses on CLIP systems is of interest to the adversarial ML community.

The results in Figure 3a are more-or-less convincing that the attacks have the intended effect.

**Weaknesses**

Besides image retrieval, zero-shot classification is perhaps the main other application of CLIP models. This paper does not examine whether their "fooling images" can successfully attack such classifiers. If these images can be detected, then I am unsure what risk they really are to real-world systems.

Generalization results in Figure 3 appear marginal: the difference in means between "Imagenet all classes" and "fooling ex. other classes" appears small (~0.01 or 0.02). Thus I am unsure how much generalization to untargeted classes is taking place.

Regarding "potential explanation" of the results in Section 4.2 that generalization occurs to "semantically related results", this could be explicitly tested by using the WordNet tree to define exactly which held-out classes are "semantically related" or not.

**Minor comments**

Some of the writing comes off as speculative and unfounded:
* "A likely explanation can be found" (Section 4.1)
* "A possible explanation could be" (Caption of Figure 3)
* "presumably due to a few mislabeled images or very hard examples" (Section 4.2)
* "which seems to imply that" (Section 4.4)

I recommended that the authors either substantiate these claims or make it crystal clear that it is just speculation.

Some of the writing is not specific enough
* "We found that... plays a central role with respect to" (Section 6).

Be more specific about what your findings are rather than use generic language like "plays a central role" .

---

> ### Author Response · Authors · 2024-02-17
> **Author response**
>
> Thank you for taking the time to review our manuscript, and for your insights and suggestions.
>
> >Besides image retrieval, zero-shot classification is perhaps the main other application of CLIP models. This paper does not examine whether their "fooling images" can successfully attack such classifiers. If these images can be detected, then I am unsure what risk they really are to real-world systems.
>
> We find that in a classification setting, where the similarity scores are passed through argmax, the attack has the same effect as a regular adversarial attack (i.e. the adversarial class with the incidentally highest similarity score becomes the predicted output class).
>
> We are unsure though what is meant here by the phrase “if these images can be detected” as we find several different things could be implied here:
>
> - Whether images with adversarial perturbations are still classified correctly: We find that that is not the case: When mined for a sufficient amount of steps, all scores of the targeted adversarial classes eventually outperform the score of the original class, which implies that one of the adversarial classes is predicted by the attacked zero-shot classifier.
>
>
> - Whether PGD-mined adversarial examples can be distinguished from the original attacked images: As the attack is intrinsically off-manifold, slight artifacts can be perceived on the images (examples of mined images can be found in Figure 6 in the Appendix). We have elaborated our views on the significance of the images being at times recognizeable as adversarial examples in Section 5.  In short, we find that even if mined images can be detected by an attentive user (a distracted user might miss the introduced artifacts of a PGD-mined image, and also a cunning attacker might hide artifacts by choosing the right template image to start from), there still remain many ways for an attacker to introduce adversarial examples where no human supervision or control is present. For instance, fooling images could be uploaded to a webservice to harm the quality of service of a productive system.
>
>
> - Whether there is a way to automatically detect CLIPMasterPrints on the basis of the adversarial images alone: We are not aware of any means to do so.
>
> Have we addressed your question as intended, and if not may we ask for clarification? Thank you.
>
>
> >Generalization results in Figure 3 appear marginal: the difference in means between "Imagenet all classes" and "fooling ex. other classes" appears small (~0.01 or 0.02). Thus I am unsure how much generalization to untargeted classes is taking place.
>
> We would like to emphasize that, while the distribution of “Imagenet all classes” here refers to the CLIP Scores of all images in the Imagenet validation set, CLIP Scores have only been computed between matched images and class-labels rather than between all images and all class-labels.
> The fact that the two distributions are, as pointed out, almost identical, therefore actually provides the evidence that CLIPMasterPrints also achieve high scores on untargeted classes.
>
> >Regarding "potential explanation" of the results in Section 4.2 that generalization occurs to "semantically related results", this could be explicitly tested by using the WordNet tree to define exactly which held-out classes are "semantically related" or not.
>
> To provide additional evidence that these untargeted classes are semantically related, we have added an additional experiment in Section 4.3. We train 5 additional CLIPMasterPrints using PGD on hyponyms (related subterms) of the four nouns “dog”, “vegetable”,”motor vehicle” and “musical instrument”. We train one CLIPMasterPrint for half of hyponyms of a particular noun and then evaluate its performance on the remaining ones. Figure 4 in the updated manuscript shows the results. As one can see, apart from a few exceptions, non-targeted classes only obtain slightly lower or at times even on par scores than the targeted classes. We therefore conclude that mined CLIPMasterPrints indeed work on semantically related, not directly targeted classes as well.
>
> > Some of the writing comes off as speculative and unfounded:
>
> We have strengthened our writing where we consider the presented evidence sufficient, as well as put emphasis on the fact that further investigation is required where ultimately delivering the required evidence would be beyond the scope of this work.
>
> >Requested Changes
>
> Thank you again for your time and insights. W.r.t requested changes, please see our responses above.

---

> > ### Comment · Reviewer_FGxL · 2024-02-21
> > **Detection of CLIPMasterPrints**
> >
> > Regarding my phrase “if these images can be detected”, I had a different interpretation in mind which I'll attempt to clarify below. I appreciate the author's comments on other possible interpretations of this phrase as well.
> >
> > Suppose that I, the maintainer of a production CLIP system, was faced with attacks on my system as described by this paper. Suppose further that I collect a dataset of such attacks and their ground-truth counterparts, and then proceed to train a new classifier, independent of the CLIP system, to detect "Is CLIPMasterPrint" vs "Not". Could I not then use such a classifier to mitigate the effect of future attacks on my system?

---

> > > ### Author Response · Authors · 2024-02-23
> > > **Thanks for clarifying**
> > >
> > > We evaluate a related mitigation strategy in the appendix (Section A.2)  where we try to refine CLIP to detect off-manifold images, and find that this does not protect against newly mined CLIPMasterPrints which are not contained in the train set.
> > > We have not tried training a different supervised classifier (e.g. a ConvNet) in the way described above yet, but investigations whether artefacts introduced by CLIPMasterPrints can be reliably detected by training such a system could be an interesting direction of future research towards mitigation strategies.

---

> > > > ### Comment · Reviewer_FGxL · 2024-02-24
> > > > **More experiments on detecting CLIPMasterPrints needed**
> > > >
> > > > As I wrote in my review, the question of whether or not CLIPMasterPrints can be reliably detected remains critical to securing my recommendation. An additional experiment with independent classifier as described above would help answer this question. The reason is that, if the attacks can be reliably detected this way (perhaps the first thing a maintainer of a production CLIP system would try), then the threat to real-world systems is low. The fact that mined images can be detected by the human eye suggests to me that classification by deep networks is indeed feasible, but I leave the final word to experiment.

---

> > > > > ### Author Response · Authors · 2024-02-27
> > > > > **Results of requested experiment and TMLR evaluation criteria**
> > > > >
> > > > > We have conducted the suggested experiment: We train a VGG19 ConvNet on a training set 50000 images (25000 of which are CLIPMasterPrints)  and validate on 50000 additional images (of which again, 50% are CLIPMasterPrints). Our preliminary results indicate that the trained system detects CLIPMasterPrints with an validation accuracy of 99.36%, which makes the proposed approach an effective mitigation strategy. After some further validation of the results, we are happy to include this experiment as an additional proposed mitigation strategy into the paper. Here it is important to note that to detect CLIPMasterPrints successfully, one has to first have a dataset of CLIPMasterPrints available to train a classifier.
> > > > >
> > > > > With respect to whether our work is a good fit for TMLR given that CLIPMasterPrints can be mitigated as proposed: we are aware that the potential threat/impact of an adversarial attack is difficult to anticipate prior to the actual attack being executed in real-world systems. Therefore, we have chosen to submit our work to TMLR as its evaluation criteria explicitly state that factual correctness and interest to the journal’s readers is favored over (anticipated) impact. We are more than happy to revise the manuscript according to the results of the above experiment to ensure factual correctness. In terms of possibly anticipated limited impact, we argue that the paper is still interesting to TMLR’s readership to raise awareness to the issue and how it is connected to the modality gap of multimodal contrastive models. In order to mount a (possibly straight-forward) defense against an attack, **one needs to be aware the attack exists.**

---

> > > > > > ### Comment · Reviewer_FGxL · 2024-02-27
> > > > > > **Questions addressed**
> > > > > >
> > > > > > Thanks for addressing my question about detection with the additional experiment. Please update the manuscript with these new results. They give a more complete picture of this work, which I believe will be valuable to the community.
> > > > > >
> > > > > > I agree that this work aligns with the editorial policies of TMLR. I will support acceptance after the next revision.

---

> > > > > > > ### Author Response · Authors · 2024-02-27
> > > > > > > **Thank you**
> > > > > > >
> > > > > > > Thank you, we will revise the manuscript as promised and upload the revision by tomorrow night (Feb 28).

---

> ### Author Response · Authors · 2024-02-28
> **Paper revised**
>
> The revised version of the paper has been uploaded.
> Kind regards,
> The authors

---

### Review · Reviewer_m6KN · 2024-02-05

**Summary Of Contributions:**

This study illustrates the vulnerability of contrastive vision-language pretraining methods to adversarial master images (CLIPMasterPrint), which are designed to maximize their similarity with any given caption. The existence of these master images is confirmed across various methodologies, including CLIP, BLIP, and SigLIP, and can be identified through different means such as SGD, LVE, or PGD.
The paper provides empirical evidence of the impact of these master images on a subset of ImageNet and a selection of painting image collections.

**Audience:**

Yes

**Broader Impact Concerns:**

No concern

**Claims And Evidence:**

Yes

**Requested Changes:**

It would be nice to clarify the behavior of CLIPMasterPrints as you increase the number classes to 1000 and beyond.

**Strengths And Weaknesses:**

Strengths:
- The paper effectively demonstrates the existence of CLIPMasterPrints.
- The study investigates several optimization procedures to identify CLIPMasterPrints.
- The research explores various CLIP methodologies (CLIP, BLIP, SigLIP).
- The paper suggests a straightforward and promising method to decrease the sensitivity to CLIPMasterPrints.

Weaknesses:
- The range of visual diversity examined in the empirical protocol is somewhat limited. The paper only considers painting images and a subset of ImageNet. It would be beneficial to confirm the existence of CLIPMasterPrints using random image/caption pairs from internet data, similar to the data used for CLIP pretraining.
- The paper demonstrates that a CLIPMasterPrint can maximize similarity with images extracted from up to 100 classes. However, it is unclear what would happen if an image is randomly selected from the full ImageNet or ImageNet-22k dataset. It remains to be seen whether a CLIPMasterPrint exists in this scenario, or if their existence is limited to specific data subsets.

---

> ### Author Response · Authors · 2024-02-17
> **Author response**
>
> Thank you for reviewing our paper and for your insights and suggestions.
>
> > The range of visual diversity examined in the empirical protocol is somewhat limited. The paper only considers painting images and a subset of ImageNet. It would be beneficial to confirm the existence of CLIPMasterPrints using random image/caption pairs from internet data, similar to the data used for CLIP pretraining.
>
> > The paper demonstrates that a CLIPMasterPrint can maximize similarity with images extracted from up to 100 classes. However, it is unclear what would happen if an image is randomly selected from the full ImageNet or ImageNet-22k dataset. It remains to be seen whether a CLIPMasterPrint exists in this scenario, or if their existence is limited to specific data subsets.
>
> While to the best of our knowledge the dataset used for CLIP pretraining is not freely available, mining CLIPMasterPrints for LAION image/caption pairs of subsets (assuming that such subsets are feasible to store and query) could be an interesting line of future work.
>
> > It would be nice to clarify the behavior of CLIPMasterPrints as you increase the number classes to 1000 and beyond.
>
> We have added Section A.3 in the appendix where we investigate the behavior of CLIPMasterPrints as we target up to 1000 classes. We find that, while performance as expected declines slightly, CLIPMasterPrints mined on all classes of the 2012 Large Scale Visual Recognition subset still outperform a significant portion of the images paired to the targeted classes in the corresponding validation set (that is, the ILSVRC 2012 validation set).
>
> Thank you again for your time and insights.

---

### Review · Reviewer_udbx · 2024-02-13

**Summary Of Contributions:**

In this paper the authors do the following:

1. Introduce a novel set of methods for acquiring adversarial "master key" images that tend to show up with very high similarity to a number of key prompts and their classes on the CLIP models.
2. Demonstrate that a concept previously dubbed the "modality gap" seems to be related to the effectiveness of these adversarial attacks, and that optimizing a model to directly reduce this gap, improves a models robustness to such attacks.
3. Show various pieces of evidence that support their claims and offer further information for exploration.
4. Demonstrate that while they learned these images on CLIP, these tend to generalize to models that have various degrees of transfer from CLIP, such as BLIP and SIGLIP.
5. Offer some practical insights into how such methods may be used to cause various degrees of problems in image retrieval systems.

**Audience:**

Yes

**Broader Impact Concerns:**

The authors cover their bases well in the paper. I would argue that this work is something that bad actors would already be thinking or doing, and such work can at least make it more clear when it happens, and stimulate more work towards defending against such attacks.

**Claims And Evidence:**

Yes

**Requested Changes:**

Narrative Related:
1. Make it more clear as to why someone should care about such adversarial attacks on multi modal models. Ideally as a paragraph in the introduction that links things to a practical application domain.
2. Clearly explain what a modality gap is in this context early on.
3. Consider rewriting bits of the paper with an assumption that most folks know how CLIP works, and they have previously studied neural networks, backpropagation and deep learning to make this paper more accessible and thus increase its impact.

Typos/Odd Phrasing Related:

In the section titled "Potential Attack Scenarios," the word "exploited" is likely meant to be "be exploited." The sentence should read, "This could be exploited in malicious ways for censorship, adversarial marketing, and disrupting the quality of service of image retrieval systems."

In the section "4.2 Generalization to semantically related prompts and labels," the term "LVE-mined image" may cause confusion. It might be clearer to specify that it refers to an image mined using the Latent Variable Evolution (LVE) technique.

In the "Broader Impact Statement," the term "adverserial" should be corrected to "adversarial." The correct sentence should read, "This could be exploited in malicious ways for censorship, adversarial marketing, and disrupting the quality of service of image retrieval systems."

Throughout the document, there's inconsistent use of "CLIPMasterPrints" and "fooling master images." For clarity and consistency, it may be beneficial to choose one term and stick with it throughout the document.

In several places, the document refers to "gradient-free approach" and "black-box approach" interchangeably when discussing the mining of fooling images without access to the model's gradients. For clarity, it would be helpful to stick to one terminology or clearly define these terms if they are intended to convey different nuances.

The document uses both "SGD" (Stochastic Gradient Descent) and "PGD" (Projected Gradient Descent) without a clear introductory explanation of these acronyms on their first use. While these are common terms in the field, a brief explanation or expansion of the acronyms at their first mention would make the text more accessible to readers not deeply familiar with these techniques.

In the appendix section titled "A.3 Analysis of information distribution in fooling master images," the sentence structure and clarity could be improved. Specifically, the last paragraph could be rewritten for clarity to better explain the significance of the findings related to the distribution of information in fooling images and their sensitivity to occlusions and cropping.

The document frequently switches between British and American spelling conventions (e.g., "optimisation" vs. "optimization"). It would be more coherent to stick to one spelling convention throughout the document.

**Strengths And Weaknesses:**

Strengths:
1. Very well written.
2. Clear evidence that supports the authors claims.
3. Nice figures and formatting throughout.
4. Interesting findings with practical applications.

Weaknesses:
1. The authors take way too long to explain why one should care about these adversarial attacks.
2. The term modality gap is mentioned a lot, and explained very little, and a bit too late. Having a succinct, but robust description of what this is would be very useful, as well as the practical details of computing it. Does one need the full ImageNet to do this? How do you backprop through all those images? etc.
3. In general there seems to be a very strong bias in the writing that the readers are very much aware of all the latest and greatest in adversarial attack land, and while that may be the case for some, rewriting bits of the paper with an assumption that most folks know how CLIP works, and they have previously studied neural networks, backpropagation and deep learning would make this paper more accessible and thus increase its impact.

---

> ### Author Response · Authors · 2024-02-17
> **Author response**
>
> Thank you for taking the time to review our paper and for your insights and suggestions.
>
> >Make it more clear as to why someone should care about such adversarial attacks on multi modal models. Ideally as a paragraph in the introduction that links things to a practical application domain.
>
> We have rewritten the corresponding paragraph in the introduction to be more specific w.r.t practical applications and what an attack could look like. Furthermore, we have added a corresponding sentence to the abstract.
>
> > Clearly explain what a modality gap is in this context early on.
>
> We have rewritten section 3 to explain the nature of the modality gap and how we exploit it more explicitly
>
> > Consider rewriting bits of the paper with an assumption that most folks know how CLIP works, and they have previously studied neural networks, backpropagation and deep learning to make this paper more accessible and thus increase its impact.
>
> Making our work as accessible as possible is very important to us. Yet we feel that for a rigorous introduction to neural networks, backpropagation, transformers and contrastive training approaches the available space of 12 pages is too limited. Furthermore, we believe that an in-depth understanding of these concepts may not be necessary for readers from other fields to grasp our work in a intuitive way.
>
> We have therefore chosen a middle path and rewritten Section 3 in order to give readers unfamiliar with the function of CLIP a high-level perspective on CLIP as mechanism to derive vector embeddings from images and text which can then be matched using a regular cosine similarity. Furthermore, we have elaborated on the details (how to compute the necessary subgradients of the loss function, the purpose of reprojecting gradients etc.) of the SGD and PGD approaches and give an high-level intuition on how these methods can be used to mine adversarial examples.
>
> > Throughout the document, there's inconsistent use of "CLIPMasterPrints" and "fooling master images." For clarity and consistency, it may be beneficial to choose one term and stick with it throughout the document.
>
> We make a distinction here between “fooling master images” as introduced by bontrager et al in general, and CLIPMasterPrints as ‘the fooling master images CLIP is vulnerable to“. We have cleaned up or writing to only introduce the general concept of fooling master images once, after which we introduce CLIPMasterPrints as the fooling master images CLIP is vulnerable to, and subsequently use the term CLIPMasterPrints in the text consistently, whenever we refer to our specific approach.
>
> > In several places, the document refers to "gradient-free approach" and "black-box appro ach" interchangeably when discussing the mining of fooling images without access to the model's gradients. For clarity, it would be helpful to stick to one terminology or clearly define these terms if they are intended to convey different nuances.
>
> We renamed the approach everywhere to “black-box approach” for simplicity and consistency
>
> >The document uses both "SGD" (Stochastic Gradient Descent) and "PGD" (Projected Gradient Descent) without a clear introductory explanation of these acronyms on their first use. While these are common terms in the field, a brief explanation or expansion of the acronyms at their first mention would make the text more accessible to readers not deeply familiar with these techniques.
>
> As stated above, we have extended the paragraphs where both methods are introduced in order to make our work more accessible to a wider audience of readers.
>
> > In the appendix section titled "A.3 Analysis of information distribution in fooling master images," the sentence structure and clarity could be improved. Specifically, the last paragraph could be rewritten for clarity to better explain the significance of the findings related to the distribution of information in fooling images and their sensitivity to occlusions and cropping.
>
> We have rewritten corresponding section in the appendix to be more clear and accessible.
>
> >Typos/Odd Phrasing Related:
> >[...]
>
> Thank you for reading our manuscript in such great detail. The stated flaws and inconsistencies have been addressed in the updated manuscript.

---

> > ### Comment · Reviewer_udbx · 2024-03-08
> > **The changes are satisfactory**
> >
> > Thank you for your hard work in addressing my comments and concerns. I believe this makes the paper a lot more water tight now! :)

---

### Author Response · Authors · 2024-02-17
**Revised paper and responses to reviewer comments**

We would like to thank all reviewers for their insightful comments, which we believe have improved our paper. The most important new additions are: (1) an experiment using WordNet hyponyms to confirm that CLIPMasterPrints in fact generalize to semantically related classes; (2) an additional section to the appendix further exploring the behavior of CLIPMasterPrints when targeting an increasing amount of classes/prompts; (3) Section 3 has been rewritten to be more accessible to a wider audience of readers. Furthermore, the connection to the modality gap and how it is exploited by CLIPMasterPrints is drawn earlier and more explicit; (4) we have elaborated our view on the fact that CLIPMasterPrint exhibit visual artifacts, and how these artifacts impact the envisioned risk for real-world systems in Section 5. (5) Formulations which have been found to be too vague by the reviewers have been revised. Furthermore a number of stylistic and typographic issues pointed out by the reviewers have been fixed.
We hope that the reviewers find these changes to be in line with their remarks, to which we respond in detail in a point-by-point fashion below.

---

> ### Author Response · Authors · 2024-02-28
> **Revision 2 Detection experiment added**
>
> Revision 2: As suggested, an experiment on automatically detecting CLIPMasterPrints by means of a classifier has been added

---

### Decision · Action_Editor_GRQ4 · 2024-03-20

**Recommendation:** Accept as is

**Comment:**

This paper identified a new limitation of CLIP models by finding fooling master images
for CLIP (CLIPMasterPrints) that are of high confidence but non-meaning to human users. The authors demonstrated the crafting of CLIPMasterPrints in the black-box setting and proposed some mitigation strategies.

This paper provides new findings to understand the limitations of CLIP models and their implications on robustness. The authors' rebuttal has addressed the main concerns of reviewers. All reviewers are in favor of accepting this submission, and I concur.

**Audience:**

Of broad interest to TMLR audience (multi-modal models, adversarial robustness)

**Claims And Evidence:**

Yes. The claims are written clearly and supported by the abundant experimental results.